# Beneficial Prognostic Effects of Aspirin in Patients Receiving Sorafenib for Hepatocellular Carcinoma: A Tale of Multiple Confounders

**DOI:** 10.3390/cancers13246376

**Published:** 2021-12-20

**Authors:** Luca Ielasi, Francesco Tovoli, Matteo Tonnini, Raffaella Tortora, Giulia Magini, Rodolfo Sacco, Tiziana Pressiani, Franco Trevisani, Vito Sansone, Giovanni Marasco, Fabio Piscaglia, Alessandro Granito

**Affiliations:** 1Division of Internal Medicine, Hepatobiliary and Immunoallergic Diseases, IRCCS Azienda Ospedaliero-Universitaria di Bologna, 40138 Bologna, Italy; luca.ielasi@studio.unibo.it (L.I.); matteo.tonnini@studio.unibo.it (M.T.); vito.sansone@unibo.it (V.S.); fabio.piscaglia@unibo.it (F.P.); alessandro.granito@unibo.it (A.G.); 2Department of Medical and Surgical Sciences, University of Bologna, 40138 Bologna, Italy; franco.trevisani@unibo.it (F.T.); giovanni.marasco@unibo.it (G.M.); 3Liver Unit, Department of Transplantation, Cardarelli Hospital, 80131 Naples, Italy; raffaella.tortora@aocardarelli.it; 4Department of Gastroenterology and Transplant Hepatology, Papa Giovanni XXIII Hospital, 24127 Bergamo, Italy; gmagini@asst.pg23.it; 5Gastroenterology Unit, Azienda Ospedaliero-Universitaria Pisana, 56126 Pisa, Italy; r.sacco@ao-pisa.toscana.it; 6Gastroenterology and Digestive Endoscopy Unit, Foggia University Hospital, 71122 Foggia, Italy; 7Humanitas Cancer Center, IRCCS Humanitas Research Hospital, Rozzano, 20089 Milan, Italy; tiziana.pressiani@cancercenter.humanitas.it; 8Semeiotica Medica, IRCCS Azienda Ospedaliero-Universitaria di Bologna, 40138 Bologna, Italy; 9Internal Medicine and Digestive Physiopathology Unit, IRCCS Azienda Ospedaliero-Universitaria di Bologna, 40138 Bologna, Italy

**Keywords:** hepatocellular carcinoma, liver cirrhosis, liver cancer, aspirin, outcome, sorafenib, systemic treatment

## Abstract

**Simple Summary:**

Low-dose aspirin has a preventive effect against multiple malignancies, including hepatocellular carcinoma. Whether aspirin could also ameliorate the prognosis of patients who already developed advanced cancer remains an elusive question. In the case of hepatocellular carcinoma, randomized clinical trials are difficult to design and observational studies suffer from multiple biases. We tried to address these problems performing a large-scale observational study, which allowed multiple corrections to reduce the main known biases. We found that among patients who received sorafenib as an antineoplastic treatment, patients who concurrently received aspirin had better outcomes than non-aspirin users. Once the confounders were properly addressed, this advantage was confirmed, albeit at the cost an increased rate of minor bleeding events. Even if we are aware that confirmatory randomized clinical trials are difficult to organize, our results warrant further investigation, as aspirin might be a low-cost, effective and safe additional treatment for selected patients with advanced hepatocellular carcinoma.

**Abstract:**

Case–control observational studies suggested that aspirin might prevent hepatocellular carcinoma (HCC) in high-risk patients, even if randomized clinical trials are lacking. Information regarding aspirin in subjects who already developed HCC, especially in its advanced stage, are scarce. While aspirin might be a low-cost option to improve the prognosis, multiple confounders and safety concerns are to be considered. In our retrospective analyses of a prospective dataset (*n* = 699), after assessing the factors associated with aspirin prescription, we applied an inverse probability treatment weight analysis to address the prescription bias. Analyses of post-sorafenib survival were also performed to reduce the influence of subsequent medications. Among the study population, 133 (19%) patients were receiving aspirin at the time of sorafenib prescription. Aspirin users had a higher platelet count and a lower prevalence of esophageal varices, macrovascular invasion, and Child–Pugh B status. The benefit of aspirin was confirmed in terms of overall survival (HR 0.702, 95% CI 0.543–0.908), progression-free survival, disease control rate (58.6 vs. 49.5%, *p* < 0.001), and post-sorafenib survival even after weighting. Minor bleeding events were more frequent in the aspirin group. Aspirin use was associated with better outcomes, even after the correction for confounders. While safety concerns arguably remain a problem, prospective trials for patients at low risk of bleeding are warranted.

## 1. Introduction

Large retrospective case–control studies suggested that patients receiving aspirin for the primary or secondary prevention of cardiovascular diseases are less likely to develop both hepatocellular carcinoma (HCC) and cholangiocarcinoma (CCA) compared to matched controls [1,2]. Nevertheless, evidence from randomized controlled trials is lacking.

Whether the hypothesized beneficial effects of aspirin are limited to the prevention of HCC occurrence, or extend to the improvement of prognosis in patients who have already developed HCC, is still matter of debate. Multimodal and sequential treatments of HCC always hindered mechanistic studies, as any potential benefit might be masked by subsequent treatments. In such a complex setting, patients undergoing systemic therapies represent an ideal population, as they are unlikely to switch back to other treatment modalities. Their prognosis is also relatively short, making the occurrence of confounding events unlikely. 

A protective role of aspirin in patients with HCC receiving sorafenib has been recently suggested [3], but many confounding factors are yet to be addressed. First, only patients without clinically relevant signs of portal hypertension (in particular esophageal varices at high risk of bleeding and severe thrombocytopenia) are usually those to whom aspirin is prescribed, thus generating a possible selection bias. Second, aspirin users are more likely to receive other drugs (in particular insulin, metformin, and statins), which have also been reported to have an impact of the survival of sorafenib-treated patients [4,5]. Third, it remains unclear whether the hypothetical benefit of the sorafenib-aspirin combination is synergistic or merely additive.

Answering all of these questions is of particular importance in the age of sequential systemic treatments [6]. After having been the mainstay of systemic treatments for HCC for a decade, sorafenib still remains a possible second-line treatment after the failure of the combination atezolizumab–bevacizumab [7].

The aim of the present study was to verify whether the use of aspirin influences prognosis in a large multicentric population of patients treated with sorafenib for advanced HCC.

## 2. Materials and Methods

### 2.1. Design of the Study

This study was performed using medical records from the archives of patients with hepatocellular carcinoma treated with sorafenib (ARPES) database. This prospective registry database was created in 2010 to collect data acquired of patients treated with sorafenib, in a real-life scenario to identify clinical, laboratory, and imaging predictors in response to the drug. This database includes consecutive patients treated with sorafenib in 5 different Italian Centers (Sant’Orsola-Malpighi Hospital, Bologna; Cardarelli Hospital, Naples; Papa Giovanni XXIII Hospital, Bergamo; Azienda Ospedaliero-Universitaria Pisana, Pisa; Humanitas Clinical and Research Center, Milan). To preserve the full originality of our data, patients added to the ARPES database by a sixth center (Meldola Hospital) were not considered for this study, as they had been previously included in a previous study of sorafenib and aspirin [3]. Data were entered every 3–6 months starting from January 2010 into electronic data files by co-investigators from each center and were checked at the data management center for internal consistency. For this study, we considered patients who were prescribed sorafenib from January 2010 to December 2019. The starting date coincided with the creation of the database and, therefore, with the possibility of obtaining prospective data from all the study centers. The closing date of the follow-up, chosen to allow an adequate follow-up of patients, was 31 December 2020.

### 2.2. Baseline Evaluation

Information about previous treatments and concurrent medications, parameters entailing the residual liver function according to the Child–Pugh score, tumor staging according to the BCLC classification, baseline α-fetoprotein (AFP) value, and performance status according to the Eastern Cooperative Group Performance Status was available for all patients. 

### 2.3. Definition of Aspirin-Treated Patients

Patients who were taking aspirin at the time of sorafenib start were considered as aspirin-treated patients, while the remaining patients were considered as non-aspirin treated patients. One patient who was not aspirin-treated at the start of sorafenib received aspirin following an arterial thromboembolic event for 46 days (from the event to the death). Conversely, two aspirin-users permanently discontinued both sorafenib and aspirin in response to severe bleeding events and died 27 and 114 days after the event.

Considering the small number of these cases and the limited impact that these cross-overs had on the prognosis, these three cases maintained their original assignation to the respective study group. 

### 2.4. Management of Sorafenib

Sorafenib was started at the usual dosage of 400 mg twice a day. Dose modifications (including dose reductions and discontinuation) were performed in cases of intolerable adverse effects. The occurrence of dermatological adverse events was recorded as an event of special interest due its known prognostic significance [8]. Sorafenib was continued until: (i) radiological and clinical progression (for patients eligible for second-line licensed drugs or clinical trials, radiological progression alone was considered sufficient for discontinuation); (ii) unacceptable toxicity; (iii) clinically significant deterioration of liver function.

### 2.5. Ethics

The study protocol was reviewed and approved by the local ethics committees. All patients gave their written informed consent for their data to be included in the prospective observational registry. The study was conducted according to the ethical guidelines of the latest Declaration of Helsinki.

### 2.6. Statistical Analysis 

Continuous variables are expressed as median and interquartile range (IQR). Categorical variables are expressed as frequencies. Group comparisons were performed with the Mann–Whitney test. Categorical variables were evaluated using the 2-tailed Fisher test. Overall survival (OS) was measured from the starting date of sorafenib therapy until the date of death or the last visit or the end of the follow-up period. Progression-free survival (PFS) was calculated from the start of sorafenib until progression or death. Post-sorafenib survival (PSS) was measured from the permanent discontinuation of sorafenib to death. Survival curves were estimated using the product-limit method of Kaplan–Meier. The role of stratification factors was analyzed with log-rank tests. To define the predictors of OS, we used a time-dependent covariates survival approach, including statistically significant clinical variables (*p* < 0.05) from the univariate Cox analysis. Remaining strictly limited to the unadjusted analysis, before propensity scores were calculated, platelet count was considered as a variable of special interest and included in the multivariable model, regardless of the univariable results (due to the very likely severe imbalance in platelet count between aspirin users and non-users). Propensity scores for receiving aspirin were calculated by performing non-parsimonious multivariate logistic regression models that included all measured potential predictors for outcome and aspirin treatment. The propensity scores were used to calculate the inverse probability of treatment weight (IPTW) in Cox analysis as a confounder. Treatment weights were calculated as 1/propensity score for patients receiving aspirin and 1/(1 − propensity score) for controls. The kernel distribution of PS for both groups was explored to ensure that a substantial part of the original population was theoretically eligible to both receive and not receive aspirin (thus indicating the applicability of results in the real-life clinical practice).

Multivariate models with backward addition of different variables to the model weighted by the propensity scores were performed, after excluding collinearity. To exclude biases deriving from outliers (particularly from cases with very low propensity scores and controls with high propensity scores) in the weighted analyses, standardized differences of all variables were used to compare the difference in means in units of the pooled standard deviation. Statistical analysis was performed using SPSS Statistics for Windows (version 24.0; IBM, Armonk, NY, USA) and STATA/SE 14.1 (StataCorp, Lakeway, TX, USA).

## 3. Results

### 3.1. Study Population

This study included all 699 patients treated with sorafenib for HCC and included in the ARPES database. Most patients were cirrhotic (93.0%). Viral hepatitis was the leading cause of chronic liver disease (hepatitis C virus infection 48.6%, hepatitis B virus infection 23.5%, nonviral causes 27.9%). Two hundred twenty-eight (32.6%) patients received post-sorafenib medications as licensed treatments, or investigational drugs, or off-label therapy. A total of 133 (19.0%) patients were on aspirin at the time of sorafenib prescription. Among them, 56 were taking aspirin following a cardiovascular event in their medical history and 77 as primary prevention. The daily doses were as follows: 75 mg (*n* = 10), 100 mg (*n* = 107), 150 mg (*n* = 3), and 160 mg (*n* = 13). 

Aspirin users significantly differed from non-users in several respects. They were more frequently male and had signs of portal hypertension less often, showing a higher platelet count and a lower prevalence of esophageal varices, macrovascular invasion and Child-Pugh B status. Notably, all varices were low risk, because patients with varices at high bleeding risk left untreated cannot be prescribed with sorafenib as per regulatory aspects. Furthermore, aspirin users were significantly more likely to receive statins (but not antidiabetic agents) compared to non-users. Instead, aspirin users and non-users did not differ in terms of age, performance status, extrahepatic spread and AFP > 400 ng/mL (Table 1).

### 3.2. Unadjusted Analysis

The crude analysis of OS showed that Child–Pugh B, performance status > 0, macrovascular invasion, extrahepatic spread, high AFP, absence of dermatological adverse events, higher platelet count, no aspirin use, and insulin use were negative prognostic factors. In a univariate subgroup analysis, the 12 patients treated with higher doses of aspirin (160 or 150 mg) had a trend toward an increased OS in comparison with the 121 patients who received lower doses (100 or 75 mg) (28.5 vs. 17.8 months, *p* = 0.083); due to the small sample size, it was not possible to perform a confirmatory multivariable analysis. 

The PFS was higher in aspirin users compared to non-users (7.9 (95% CI 5.3–10.4) vs. 4.3 (3.9–4.7) months, *p* < 0.001).

The disease control rate was 66.9 and 48.8% in aspirin users and non-users, respectively (odds ratio 0.471, 95% CI 0.316–0.700, *p* < 0.001). Sorafenib was permanently discontinued in 684 (97.9%) patients, due to disease progression (64.3%), unmanageable adverse events (21.3%), or liver failure (14.3%). The prevalence of aspirin users was similar among these three categories (18.0 vs. 21.2 vs. 13.3%, respectively; *p* = 0.284). The post-sorafenib survival was significantly longer in aspirin users (median 6.8 vs. 4.5 months, *p* = 0.011). A multivariable Cox regression model, including both aspirin use and known predictors of post-sorafenib survival (ECOG-PS, extrahepatic tumor spread, macrovascular invasion, and reason for sorafenib discontinuation), showed that aspirin use was an independent predictor of post-sorafenib survival (HR 0.769, 95% CI 0.620–0.954, *p* = 0.017).

There were no differences in bleeding events between aspirin users and non-users as a whole (12.0 vs. 11.5%, *p* = 0.880), or stratified as gastrointestinal (10.5 vs. 9.0%, *p* = 0.619), and non-gastrointestinal (3.0 vs. 2.7%, *p* = 0.770) bleeding. 

### 3.3. Propensity Score Analysis

Propensity scores were calculated for all patients, considering all differences between aspirin users and non-users. The distribution of probability weights is reported in Figure 1. 

After applying weights, the two populations did not differ significantly in key baseline parameters (Table 2). Evolutive events potentially affecting OS (i.e., dermatological adverse events) also did not differ in the groups after weighting (standardized mean difference 8.8%).

### 3.4. Adjusted Analyses

The adjusted median OS was 23.8 vs. 12.4 months (Figure 2). 

The weighted Cox regression confirmed independent predictors of survival aspirin use (HR 0.685, 95% CI 0.529–0.888), platelet count (HR 1.002, 95% CI 1.001–1.003), and esophageal varices (1.223, 95% CI 1.003–1.518) (Table 3). 

Aspirin users were also confirmed to have a higher disease control rate (57.7 vs. 49.4%, *p* = 0.002) and longer progression free survival (8.5 (3.4–17.0) vs. 4.3 (2.7–8.0) months, *p* < 0.001) and post-sorafenib survival (6.8 vs. 4.1 months, *p* = 0.007).

Overall bleeding events were more frequent in aspirin-treated than in non-aspirin-treated patients (18.1 vs. 11.3%, *p* < 0.001). This finding was due to an increase in both gastrointestinal (14.4 vs. 9.0%, *p* = 0.002) and non-gastrointestinal (6.4 vs. 2.6%, *p* = 0.001) events. However, the rate of grade > 3 bleeding events was similar across the study groups (4.6 vs. 3.9%, *p* = 0.401) (Figure 3). 

## 4. Discussion

In this retrospective study, aspirin use was associated with longer survival in patients with unresectable HCC not amenable to locoregional procedures and treated with sorafenib. Aspirin might prevent the development of HCC in at-risk patients [2,9,10], although its impact on the prognosis of patients who already developed HCC remains undefined. In this respect, patients receiving curative procedures have never been investigated, likely due to the fact that these therapies eradicate the target tumor and its recurrence is usually bound to either a de novo HCC or metastatic lesions. A recent retrospective study reported that patients undergoing transarterial embolization, who were receiving aspirin due to cardiovascular diseases, had a substantially better survival compared to those not receiving aspirin (HR 0.498; CI 95% 0.280–0.888; *p* < 0.018) [11]. To the best of our knowledge, only one study investigated the effect of aspirin in the setting of systemic treatments of HCC. Casadei-Gardini et al. [3] evaluated 304 patients on sorafenib, reporting that aspirin users had better OS and PFS compared to non-users. Unfortunately, due to the low sample size, the authors were not able to address some confounders (such as concurrent drugs and post-sorafenib treatments). Moreover, this result has not received an external validation so far. Our results do not merely validate these initial findings in a larger series; they also strengthen the quality of evidence (even if by means of an observational study). In fact, we checked for the multiple confounders that might affect the prescription of aspirin and evaluated the possible interference of concurrent medications, including statin and antidiabetic drugs. Moreover, we analyzed outcomes of patients after the sorafenib discontinuation, considering all known possible confounder factors that could differ between the two cohorts. Notably, the longer post-sorafenib survival we found in the aspirin-users group suggests that aspirin exerts a beneficial effect which is, at least in part, independent from that of sorafenib. 

Lastly, we provided information on the safety of aspirin in the fragile setting of cirrhotic patients receiving sorafenib. Once confounders were considered, the concurrent aspirin treatment was associated with a slight but statistically significant increase of the overall bleeding risk. However, the rate of severe (>grade 3) bleedings was not increased, in line with previous data obtained in patients on aspirin for the prevention of gastrointestinal cancers [12,13]. 

Given that aspirin inhibits platelet aggregation, we explored the intertwined role of platelets and aspirin in our HCC patients treated with sorafenib. Platelet count was inversely correlated with OS, and this result was confirmed after adjusting for potential confounders, such as signs of portal hypertension, concurrent drugs, and demographics. The adverse prognostic effect on platelets would appear in contrast with the fact that thrombocytopenia is usually associated with advanced liver disease and clinically significant portal hypertension [14]. However, some authors have already reported an association between higher platelet count and an aggressive HCC phenotype [15,16,17]. Moreover, both high platelet count and platelet-to-lymphocyte ratio have been reported as negative prognostic factors in HCC patients [18]. The in vitro relationship between platelets and tumor microenvironment is complex [19,20]. Indeed, tumor cells enhance platelet activity through an increased adenosine diphosphate secretion, and activated platelets, in turn, release a vascular endothelial growth factor (VEGF), thus promoting neoangiogenesis and cancer cell proliferation [21,22,23,24]. Moreover, other in vitro studies showed that several platelet-derived growth factors (IGF-1, HGF, TGFβ, VEGF, PDGFβ, FGF, EGF, and serotonin) can blunt the action of anti-angiogenetic drugs, such as sorafenib and regorafenib [25,26]. Interestingly, aspirin and gut microbiota can interact by exerting protection against colorectal cancer [27]. While no studies regarding similar mechanisms in liver neoplasms have been concluded, such additional interaction cannot be fully excluded.

Our study has some limitations deserving discussion. First, although based on consecutive data and adjusted for potential confounding factors, our analyses are retrospective in nature and hence vulnerable to unintended biases. Prospective randomized clinical trials are required to generate the highest quality of evidence. Nevertheless, a retrospective observational nature is typical of seminal exploratory studies. Despite addressing many of the evaluable confounders, the possibility of other unknown confounders still exists. Among these confounders, cardiovascular comorbidities are the most obvious. However, the presence of cardiovascular comorbidities is more likely to impair the survival of aspirin users, rather than improving it. Therefore, it could lead to an underestimate and not an overestimate of aspirin benefit. Moreover, misclassification of patients due to the possible use of over-the-counter aspirin of patients registered as non-aspirin users, or to the lack of adherence of aspirin users, would lead to a similar underestimate of aspirin benefits [28]. Second, only a relatively small proportion of patients received second-line treatments which, in addiction, were not homogeneous. Therefore, reliable analyses about the role of aspirin in patients receiving other tyrosine kinase inhibitors were not feasible, particularly regarding the sorafenib–regorafenib and sorafenib–cabozantinib sequences). On the other hand, the small number of patients undergoing second-line therapy limited the risk of biases due to the administration of post-sorafenib treatments. Finally, the sample size of our study prevented us from demonstrating a dose–response relationship in a multivariable model (an event which would have strengthened our hypothesis), even if a trend was found in the univariate analysis.

## 5. Conclusions

In conclusion, patients with advanced HCC who received aspirin during sorafenib treatment had better outcomes in terms of radiological response, PFS, post-sorafenib survival, and OS compared to the remaining patients. This advantage persisted after adjustment for confounders and despite an increased rate of minor bleeding events. Further studies might elucidate whether our results can be extended to patients receiving other TKIs, immune checkpoint inhibitors, either a single-agent, or in combination with other agents. The increased risk of bleeding, however, might be of concern not only for patients receiving tyrosine kinase inhibitors, but also for those who will receive a combination including bevacizumab, given the known association between this agent and hemorrhaging events. These concerns might hinder prospective clinical trials in patients with advanced HCC in a scenario already characterized by manifold issues [29], including little appetite from the pharmaceutical industry in supporting an out-of-patent cheap medication and the need for a large population to correctly assess all of the possible safety issues.

## Figures and Tables

**Figure 1 cancers-13-06376-f001:**
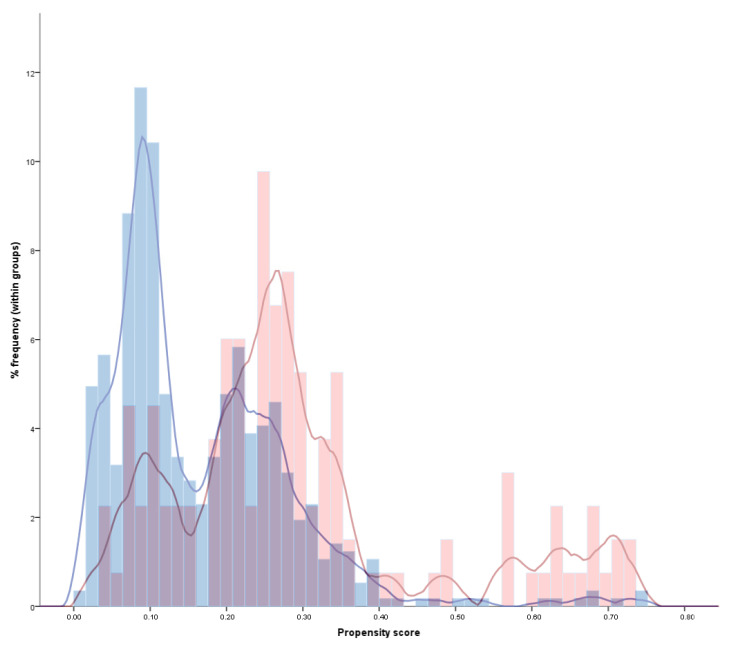
Histogram and kernel distribution of the propensity scores in aspirin-treated (red) and non-aspirin-treated (blue) patients. This figure illustrates how a non-negligible portion of patients was theoretically eligible to both receive and not receive aspirin (overlapping purple areas), thus indicating the applicability of results in the real-life clinical practice. At the same time, a relevant portion of aspirin and non-aspirin treated patients had non-overlapping scores, confirming that a propensity score analysis was needed to address the intrinsic differences between the two groups and avoid a relevant prescription bias.

**Figure 2 cancers-13-06376-f002:**
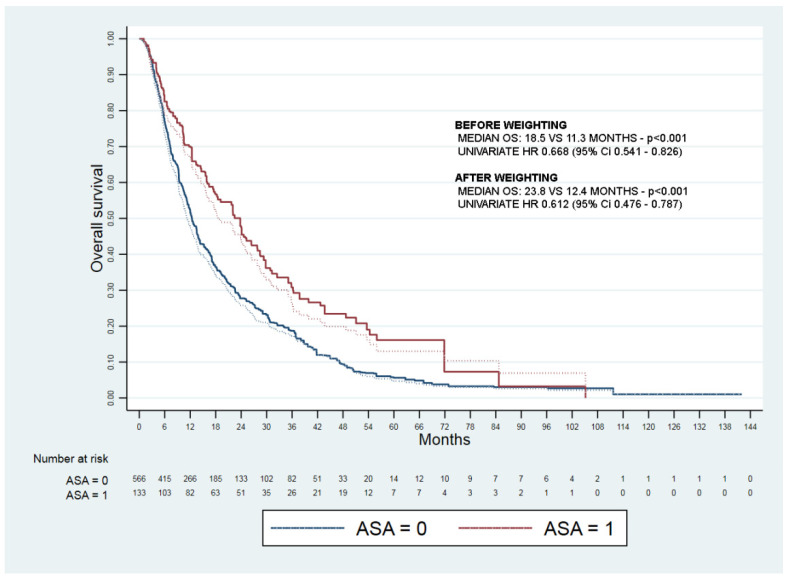
Kaplan–Meyer curves of overall survival before (dashed lines) and after weighting (full lines) of aspirin treated (ASA = 1) and not treated (ASA = 0) patients.

**Figure 3 cancers-13-06376-f003:**
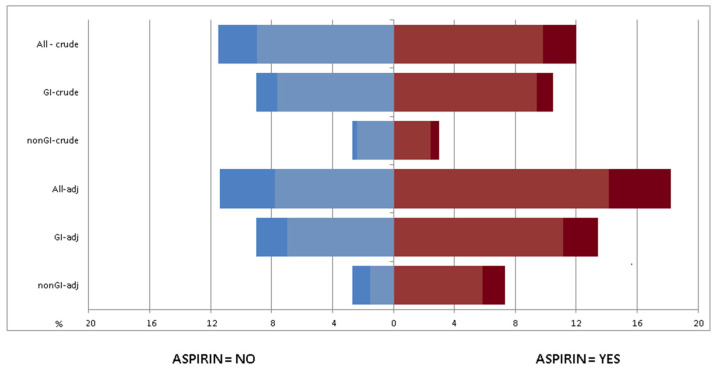
Rate of total, gastrointestinal (GI), and non-gastrointestinal (nGI) bleeding events in the crude and adjusted (adj) analysis. Dark colors indicate grade ≥ 3 events.

**Table 1 cancers-13-06376-t001:** Comparison of the characteristics of aspirin users and non-users.

Variables	All Patients *n* = 699	ASA Users *n* = 133	Non-Users *n* = 566	*p*
Age (yrs)	68 (60–74)	72 (67–79)	68 (64–76)	0.444
Sex (male)	596 (85.3)	126 (93.3)	470 (83.3)	0.003
Viral etiology	503 (72.0)	81 (60.0)	422 (74.8)	0.001
Disease duration (months)	14.3	12.7	14.7	0.215
Previous HCC treatments	518 (74.1)	96 (72.1)	422 (74.3)	0.721
Bilirubin (mg/dL)	0.93 (0.68–1.40)	0.92 (0.60–1.31)	0.95 (0.62–1.30)	0.884
Albumin (g/dL)	36 (33–39)	37 (34–40)	36 (32–39)	0.482
Platelets (10^3^/mL)	135 (95–186)	156 (115–200)	125 (92–182)	0.001
ALBI grade 1	131 (18.7)	39 (28.9)	92 (16.3)	0.001
Child-Pugh B	41 (5.9)	2 (1.5)	39 (6.9)	0.013
Esophageal varices	300 (42.9)	28 (20.7)	272 (48.2)	<0.001
Metformin	89 (12.7)	19 (14.1)	70 (12.4)	0.569
Insulin	80 (11.4)	16 (11.9)	64 (11.3)	0.881
Statin	45 (6.4)	25 (18.5)	20 (3.5)	<0.001
AFP > 400 ng/mL	217 (31.0)	34 (25.2)	183 (32.4)	0.120
ECOG-PS > 0	150 (21.5)	28 (20.7)	122 (21.6)	0.907
Tumor > 50% of liver volume or main trunk PVT	29 (4.1)	5 (3.7)	24 (4.3)	1.000
Macrovascular invasion	275 (39.3)	43 (31.9)	232 (41.1)	0.050
Extrahepatic spread	220 (31.5)	51 (37.8)	169 (30.0)	0.081
BCLC-intermediate stage	191 (27.3)	39 (28.9)	152 (27.0)	0.668

HCC: hepatocellular carcinoma; AFP: alfa-fetoprotein; ECOG-PS: Eastern Cooperative Oncology Group-Performance Status; BCLC: Barcelona Clinic for Liver Cancer.

**Table 2 cancers-13-06376-t002:** Post-weighting standardized mean differences in study variables. Continuous variables are expressed as percentages (including standardized differences).

Variable	Aspirin Users	Non-Users	Standardized Difference
Age (yrs)	67.0	67.0	+0.3
Sex (male)	83	85	−5.6
Viral etiology	73	71	+4.4
Esophageal varices	47	43	+8.0
Platelets (×10^3^/mmc)	148.1	149.9	−2.2
ALBI grade 1	19	19	+0.9
Child-Pugh B	7	6	+4.3
ECOG-PS > 0	21	21	−0.8
Tumor > 50% of liver volume or main trunk PVT	3	4	−6.4
Macrovascular invasion	38	39	−2.0
Extrahepatic spread	37	33	+8.8
AFP > 400 ng/mL	32	31	+2.2
Metformin	12	14	−5.7
Insulin	13	11	+2.9
Statin	6	7	−3.9

ECOG-PS: Eastern Cooperative Oncology Group-Performance Status; BCLC: Barcelona Clinic for Liver Cancer; AFP: alfa-fetoprotein.

**Table 3 cancers-13-06376-t003:** Weighted multivariable Cox regression of overall survival.

Variable	Univariable Analysis			Multivariable Analysis		
	HR	95% CI	*p*	HR	95% CI	*p*
Age (yrs)	1.006	0.997–1.070	0.197	-	-	-
Sex (male)	0.876	0.593–1.294	0.506	-	-	-
Viral etiology	1.195	0.959–1.490	0.112	-	-	-
Platelets (10^3^/mmc)	1.001	1.000–1.002	0.062	1.002	1.001–1.003	0.001
Aspirin	0.612	0.476–0.787	<0.001	0.685	0.529–0.888	0.004
Metformin	0.912	0.663–1.257	0.575	-	-	-
Insulin	1.438	1.137–1.820	0.002	1.277	0.964–1.691	0.088
Statin	0.747	0.524–1.044	0.087	0.815	0.590–1.126	0.216
Varices	1.363	1.113–1.668	0.003	1.223	1.002–1.518	0.048
ALBI grade 1	0.806	0.626–1.038	0.095	0.951	0.738–1.228	0.703
Child-Pugh B	4.069	2.907–5.700	<0.001	3.103	1.873–5.140	<0.001
AFP > 400 ng/mL	1.511	1.162–1.966	0.002	1.520	1.204–1.918	<0.001
ECOG-PS > 0	1.706	1.300–2.241	<0.001	1.510	1.164–1.961	0.002
Tumor > 50% of liver volume or main trunk PVT	2.798	1.935–4.047	<0.001	1.634	1.012–2.639	0.044
Macrovascular invasion	1.587	1.286–1.960	<0.001	1.558	1.245–1.949	<0.001
Extrahepatic spread	1.209	0.997–1.578	0.097	1.360	1.069–1.731	0.012
Dermatological AEs *	0.680	0.551–0.838	<0.001	0.679	0.558–0.826	<0.001

AFP: alfa-fetoprotein; ECOG-PS: Eastern Cooperative Oncology Group-Performance Status; PVT: portal vein thrombosis; AEs: adverse events. * Evaluated as a time-dependent variable.

## Data Availability

The data presented in this study are available on request from the corresponding author. The data are not publicly available due to privacy restrictions.

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
