# Peer review of "Beneficial Prognostic Effects of Aspirin in Patients Receiving Sorafenib for Hepatocellular Carcinoma: A Tale of Multiple Confounders"

_cancers, 2021, doi:10.3390/cancers13246376_

Round 1
Reviewer 1 Report
The authors adequately revised the manuscript.
I only have one suggestion left:
Line 55. "Cholaniocellular carcinoma" is not a correct term. Use cholangiocarcinoma instead.
Author Response
Thanks for pointing out this typo. The error has been corrected
Reviewer 2 Report
No further questions
Author Response
Thank your for having appreciated our manuscript
Reviewer 3 Report
This manuscript was revised well.
Author Response
Thank your for having appreciated our manuscript
This manuscript is a resubmission of an earlier submission. The following is a list of the peer review reports and author responses from that submission.
Round 1
Reviewer 1 Report
The authors conducted a study to explore the associations between aspirin use and the prognosis of patients who received sorafenib for advanced HCC. Although it is an interesting topic, the authors should be cautious of the healthy user effect and be more conservative about their findings.
Detailed comments:
- In the abstract and the introduction, the authors stated that aspirin can prevent HCC as a fact. However, there are no randomized prospective studies proving that. All those case control studies can only produce hypothesis.
- The definition of aspirin user is too simple. Currently, patients who were taking aspirin at the time of sorafenib start were considered as aspirin users. How many of them continued to do so during the sorafenib treatment course? Did any patients start aspirin after the initiation of sorafenib treatment?
- The authors should calculate the aspirin exposure during sorafenib treatment to examine if there is dose response relationship. If there is, the association between aspirin use and survival will be more convincing.
- Lines 159-160. The aspirin dose of only 123 patients were not reported, why?
- Table 1. Please add a column of all patients, so readers could easily know the characteristics of entire population. Also, please add a row listing the total number of each population for easier understanding.
- Because the majority of patients had Child A liver function, please also report ALBI and adjust it in the multivariate analysis and the propensity score.
- CLIP score has been repeatedly demonstrated to predict survival of patients with advanced HCC. Please also adjust CLIP scores (or at least the missing component: main portal vein thrombosis and liver involvement > 50%" in the multivariate analysis and the propensity score.
- Line 177. "Conversely, aspirin use was an independent predictor of longer OSl (Table 2)" I don't know where such a description came from, especially in a section entitled unadjusted analysis. Also, Table 2 does not seem to associated with this description.
- Line 184. "The multivariable logistic regression model showed that aspirin use was an independent predictor of post-sorafenib survival. Why was logistic regression used instead of Cox proportional hazard model for a time dependent variables? Which variables were put into the model and how were the variables selected?
- Tables 2 and 3. I don't think it is adequate to include dermatologic AEs in the analysis because it is a posttreatment event.
Reviewer 2 Report
In this article, the author tried to evaluate the effects aspirin in hepatocellular carcinoma patients received sorafenib. Although some paper indicated that aspirin use may improve the clinical outcome of patients with advanced hepatocellular carcinoma receiving not only sorafenib but also regorafenib. But the author tried to verify whether the use of aspirin influences prognosis in a large multicentric population of patients treated with sorafenib for advanced HCC. The idea is good and results are well organized. However, some points need to be addressed.
Major Points:
- The authors used the same hEpatocellular carcinoma treated with Sorafenib (ARPES) database with another paper they published in Clin Transl Gastroenterol (2021 Jan; 12(1)). In that paper, there are 6 different Italian Centers in the database. But in this article, the authors only mentioned about 5 different Italian Centers. Please explain the reason.
- In the 3.1 section, the author mentioned that “The daily doses were as follows: 75 mg (n=10), 100 mg (n=97), 150 mg (n=3), and 160 mg (n=13)”. Did the different doses affect the effect of aspirin treated with sorafenib for advanced HCC.
- In figure 1, please describe in detail with the result.
- In figure 2, please define the ASA=0 and ASA=1.
- Where is the statistical difference in figure 2. Please indicate the p value and HR in this result.
- There is a typo error in line 33 “carcinoma..”. Please correct it.
Reviewer 3 Report
The present study reports an incoming issue with controversial results found in the already published studies. This article considered the concerns derived from multiple confounders and safety, as well as presents interesting results from a retrospective analysis employing aspirin in sorafenib-treated patients with HCC. Despite the high number of studies that evaluate effects and benefits derived from aspirin treatment in HCC patient outcomes treated with sorafenib, lack of investigations that fulfill a retrospective study including multiple data sources for improving reliability of results makes this article a novel study for the scientific community. Authors employed patient data from different centers, properly described the methodology used for the analysis and presented the results in a well-organized structure that provide a good quality for this research.
Reviewer 4 Report
Author showed that beneficial prognostic effects of aspirin in patients receiving sorafenib for hepatocellular carcinoma.
I understood that OS was improved by aspirin in patients receiving sorafenib for hepatocellular carcinoma.
However, it is not uncertain in this rationale.
Is it the response rate and PFS?
Author should show that the response rate or PFS was improved by aspirin in patients receiving sorafenib for hepatocellular carcinoma.
